# Multi-task lesion segmentation with a lightweight U$^2$-Net to enhance explainability of mobile screening systems for diabetic retinopathy

**Marlin Siebert**[1]                                   M.SIEBERT@UNI-LUEBECK.DE

**Philipp Rostalski**[1]                           PHILIPP.ROSTALSKI@UNI-LUEBECK.DE

[1] *IME, Universität zu Lübeck, Moislinger Allee 53-55, 23558 Lübeck, Germany*

## Abstract

In addition to the recent development of deep learning-based, automatic detection systems for diabetic retinopathy (DR), efforts are being made to improve the explainability of those systems, which are usually designed as black-box models. By providing precise segmentation masks for lesions being related to the severity of DR, a good intuition about the reasoning of the diagnosing system can be given. Additionally to this progress, the development of light-weight, smartphone-based DR detections systems, being enabled by the growing computing power of edge devices, is of increasing interest in the research community. Currently, however, only very few diagnosing systems are pursuing both: implementing joint lesion segmentation and disease grading as well as using small, efficient architectures allowing for implementation on edge devices. In this paper, we evaluate the performance of a lightweight network implementation for lesion segmentation and assess its potential to extend mobile DR-grading systems for improved reasoning. To this end, the performance of a downscaled U$^2$-Net, a recent derivative of the well-known U-Net, is evaluated and compared in single- and multi-task lesion segmentation to further reduce memory cost from saving redundant models. Experimental results show promising diagnostic accuracy while maintaining a small memory footprint as well as reasonable inference speed and thus indicate a promising first step towards mobile diagnostic being able to provide both precise lesion segmentation and DR-grading.

**Keywords:** diabetic retinopathy, deep learning, multi-lesion segmentation, u-net, fundus image, mobile diagnostic

## 1. Introduction

With the increase in performance of deep vision recognition systems, developing deep learning supported diabetic retinopathy (DR) detection systems has gained significant attention. Nevertheless. the black-box character of deep learning models lack understandable reasoning of the predicted DR grade. Hence the focus is recently shifting towards providing precise segmentations of pathologic lesions related to DR along with the predicted DR grade. To solve the difficult task of providing precise segmentations, highly specialized and usually large and deep networks are developed. However, in context of smartphone-based detection systems, the implemented network architectures are required to have low memory consumption and to be computationally efficient while maintaining high diagnostic performance. In this paper, we, evaluate the impact of both network size and the use of multi-task setups to further reduce memory consumption on the models' segmentation performance using the U$^2$-Net (Qin et al., 2020) implementing a nested U-Net-structure.

## 2. Image Data and Preprocessing

For training, the Indian Diabetic Retinopathy Image Dataset (IDRiD) (Porwal et al., 2018) is used following the proposed data split into 54 training and 27 test colour fundus images with fine-grained lesion segmentation masks for microaneurysm (MA), haemorrhages (HE), hard- (HX) and soft exudates (SX). As preprocessing, the images are cropped to the visible retinal disc, resized to $s = (512, 512)$ pixels, contrast-enhanced using CLAHE (contrast limited adaptive histogram equalization) and normalized with mean and standard deviation of the database. During training, the images are also randomly augmented with a maximum rotation of 90°, an horizontal and vertical shift of $\pm(0.1s)$, using affine scaling in the range of $\pm(0.4s)$ and random horizontal and vertical flipping with a probability of 50 % is applied.

## 3. Experimental Setup

The number of features used to initialize the downscaled $U^2$-Net was experimentally determined by decreasing the feature depth until the network does no longer learn from the training data. This $U^2$-XS-Net counts only 0.29 M parameters and achieves an inference speed of about 44 fps. As a baseline, the original sized $U^2$-Net with 43.87 M parameters is trained three times in single-task (ST) mode alongside the $U^2$-XS-Net, which is additionally trained in dual- (DT) and multi-task (MT) mode. In DT mode two separate models are used that each output two segmentation masks for either red lesions (MA, HE) or white lesions (HX, SX) while in MT mode only a single network is used. Training was conducted on an Nvidia GeForce RTX 2080Ti over 400 epochs using Adam with learning rate set to $1e-3$, L2-norm weight decay set to $1e-4$ and a step scheduler decreasing the learning rate beginning from the 115th epoch on with a rate of 0.75 every 15 epochs. The loss was chosen as a weighted fusion of focal binary cross entropy (fBCE) and dice loss (D) according to

$$L(y, \hat{y}) = \frac{1}{\mathcal{C}} \sum_{k}^{\mathcal{C}} \left( \alpha \cdot \text{fBCE}_k(y_k, \hat{y}_k, \gamma) + (1 - \alpha) \cdot \text{D}_k(y_k, \hat{y}_k) \right) \cdot w_k(y_k) \tag{1}$$

with $w_k(y_k) = \left( \frac{\sum_n (1 - y_{n,k})}{\sum_n (y_{n,k}) + 1} \right)^{1/8} \cdot \left( \sum_k^{\mathcal{C}} \left( \frac{\sum_n (1 - y_{n,k})}{\sum_n (y_{n,k}) + 1} \right)^{1/8} \right)^{-1}$, $\alpha = 0.75$, $\gamma = 1.0$. The loss is applied channel-wise to the sigmoid activated output of the network and deep supervision with equally weighted errors of all side outputs of the $U^2$-XS-Net is used.

## 4. Results and Discussion

From the results displayed in Table 1 it can be seen that the $U^2$-XS-Net achieves similar performance as the baseline. When used in DT mode, the $U^2$-XS-Net outperforms the other modes except for the HX segmentation that performed best in ST mode. Compared to the listed results from literature, both estimated to use larger models as the $U^2$-XS-Net, also reasonable performance is achieved except for the MA detection which is a difficult task even for experienced physicians. Nevertheless, using the $U^2$-Net as well as the $U^2$-XS-Net with the above-described training setup slightly improves the performance for SX segmentation. In comparison to Guo et al. (2019b) who similarly proposed a lightweight network for HX segmentation with 1.9 M parameters running at 11.1 fps with input of size $(1440 \times 960)$ and

Table 1: The results are given as area under PR-curve (AUPR) with the standard deviation enclosed in brackets and the mean AUPR ($\overline{\text{AUPR}}$) across the different lesions.

| Model | MA | HE | HX | SX | $\overline{\text{AUPR}}$ |
|---|---|---|---|---|---|
| (Guo et al., 2019a) | 0.46 | 0.64 | 0.80 | **0.71** | 0.65 |
| (Yan et al., 2019) | **0.53** | **0.70** | **0.89** | 0.68 | **0.70** |
| U²-Net (ST) | **0.42** ($< 0.01$) | 0.68 ($< 0.01$) | **0.87** (0.01) | **0.74** (0.01) | **0.68** |
| U²-XS-Net (ST) | 0.41 (0.01) | 0.68 (0.01) | 0.87 (0.01) | 0.68 (0.04) | 0.66 |
| U²-XS-Net (DT) | 0.41 (0.01) | **0.69** (0.02) | 0.83 (0.02) | 0.74 (0.02) | 0.67 |
| U²-XS-Net (MT) | 0.36 (0.01) | 0.62 (0.02) | 0.85 (0.01) | 0.74 (0.01) | 0.64 |

achieving a dice score of 0.7815, the U²-XS-Net is more than six times smaller, reaches a similar dice score of 0.7967 in ST mode and is significantly faster due to the reduced input size. With these results, using the U²-XS-Net is a promising design decision for precise, mobile lesion segmentation in particular with using the DT mode. Improving performance, specifically for MA segmentation, while maintaining a small memory footprint and further decreasing the computational load will be subject to future research as well as fusing the proposed lesion segmentation with a mobile applicable DR grading system.

## Acknowledgments

This work is supported by the *Joachim Herz Stiftung* as part of the PASBADIA project.

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
