# OpenReview forum: "Multi-task lesion segmentation with a lightweight U$^2$-Net to enhance explainability of mobile screening systems for diabetic retinopathy"
_MIDL.io/2021/Conference/Short — Submitted to MIDL 2021_

### Official Review · Reviewer_o1tk · 2021-04-25

**Confidence:** 5
**Final Rating:** 2

**Summary:**

This paper investigated a simplified U2-Net for efficient retinal image segmentation. The authors reduced the feature numbers of U2-Net and used the IDRiD dataset for experiment. The result showed that the smaller model size only led to slight performance drop. The method seems to be too simple and the authors did not consider other techniques for reducing the model size. They did not even report the model size and inference time for the compared networks.

**Strengths:**

1. Lightweight structure is of interest for medical image analysis when used on edge devices. The topic of this work is interesting.
2. The authors used public dataset for evaluation.
3. The method is easy to understand.

**Weaknesses:**

1. The authors mentioned explainability in the title, abstract and introduction, but did not show how to achieve explainability in the method and results. The organization of this paper seems to be poor.
2. Reducing the channel number is a naive implementation. There have been many other methods to achieve lightweight networks, such as depth-separable convolution. The authors did not provide any comments or experimental results for these works.
3. This paper investigates lightweight models, but did not report the model size and inference time for different methods.


**Deanonymize Review:**

yes

**Detailed Comments:**

As mentioned above, this paper is not well organized. Explainability seems to irrelevant to this work. The authors should either remove these descriptions or add some results for the explainability. For lightweight structures, model size and inference time are key metrics for evaluation, but the authors failed to provide the values for these metrics. It would be good to comparing other techniques to reduce the model size, rather than simply reducing the channel numbers.

**Justification Of The Rating:**

"explainability" in the tittle is not related to the method/results of this paper. Also the experimental setting is too simple and the authors did not report the model size, despite that this paper tries to investigate a lightweight segmentation model.

**Paper Type:**

validation/application paper

**Special Issue:**

no

---

### Official Review · Reviewer_crjS · 2021-04-28

**Confidence:** 4
**Final Rating:** 1

**Summary:**

This short paper proposes to use a lightweight version of Unet to segment diabetic retinopathy lesions in retinal fundus images. For the said lightweight Unet, two networks were trained 1 segments red lesions, 1 white lesions. Another lightweight Unet is trained to segment different lesion type simultaneously.

**Strengths:**

The paper focuses on a very important topic, which can revolutionize healthcare in countries where access to clinics is difficult, therefore the availability of phone based DR classification is key. I encourage the author to continue put effort in the problem.

**Weaknesses:**

I found following the paper challenging and the description single task of the standard Unet not really clear.  I have the impression that the test set was used during training to monitor the convergence of the models, this is not good practice and should be avoided to prevent overly optimistic performance. Usually in segmentation problem dice scores are also provided.

**Deanonymize Review:**

no

**Detailed Comments:**


If I understood correctly, dual task and multitask are more single or multi class segmentation. As a consequence wrt the multitask case, it is not clear why the sigmoid activation function was used at the prediction step, aren't the segmented classes mutually exclusive here?
With respect to grading, which is one of the goal pursued according to the abstract, at which step the DR is graded? How accurate is the grading? I don't think this has been reported.


**Justification Of The Rating:**

For all the motivations above, I don't think this paper is ready for presentation at a venue like midl, even if this is a short paper, it is not convincing, especially if no portion of the data was held out for testing.

**Paper Type:**

validation/application paper

**Special Issue:**

no

---

### Meta-Review · Program_Chairs · 2021-05-10

**Recommendation:** Reject
**Confidence:** 4

**Metareview:**

Both reviewers point out important shortcomings of the paper related to clarity, experimental setup, and presentation of the results.

---

### Decision · Program_Chairs · 2021-05-11

Reject